# Membrane Chromatography-Based Downstream Processing for Cell-Culture Produced Influenza Vaccines

**DOI:** 10.3390/vaccines10081310

**Published:** 2022-08-13

**Authors:** Zeyu Yang, Xingge Xu, Cristina A. T. Silva, Omar Farnos, Alina Venereo-Sanchez, Cécile Toussaint, Shantoshini Dash, Irene González-Domínguez, Alice Bernier, Olivier Henry, Amine Kamen

**Affiliations:** 1Viral Vectors and Vaccines Bioprocessing Group, Department of Bioengineering, McGill University, Montreal, QC H3A 0G4, Canada; 2Department of Chemical Engineering, Polytechnique Montreal, Montreal, QC H3T 1J4, Canada

**Keywords:** influenza strains, H1N1, H3N2, and H7N9, cell-culture derived influenza vaccine, downstream process, membrane-based chromatography

## Abstract

New influenza strains are constantly emerging, causing seasonal epidemics and raising concerns to the risk of a new global pandemic. Since vaccination is an effective method to prevent the spread of the disease and reduce its severity, the development of robust bioprocesses for producing pandemic influenza vaccines is exceptionally important. Herein, a membrane chromatography-based downstream processing platform with a demonstrated industrial application potential was established. Cell culture-derived influenza virus H1N1/A/PR/8/34 was harvested from benchtop bioreactor cultures. For the clarification of the cell culture broth, a depth filtration was selected as an alternative to centrifugation. After inactivation, an anion exchange chromatography membrane was used for viral capture and further processing. Additionally, two pandemic influenza virus strains, the H7N9 subtype of the A/Anhui/1/2013 and H3N2/A/Hong Kong/8/64, were successfully processed through similar downstream process steps establishing optimized process parameters. Overall, 41.3–62.5% viral recovery was achieved, with the removal of 86.3–96.5% host cell DNA and 95.5–99.7% of proteins. The proposed membrane chromatography purification is a scalable and generic method for the processing of different influenza strains and is a promising alternative to the current industrial purification of influenza vaccines based on ultracentrifugation methodologies.

## 1. Introduction

The emergence of new influenza viruses from zoonotic origins and the threat of a new global pandemic have been and continue to be of great concern for the World Health Organization (WHO) [1]. Although difficult to estimate, annual epidemics of influenza are thought to result in between three and five million cases of severe illness and between 290,000 and 650,000 deaths every year around the world [2]. Influenza virus is an enveloped virus belonging to the family of *Orthomyxoviridae* [3]. While four types of antigenically distinct influenza viruses are described, only influenza A and B viruses (IAV and IBV, respectively) are responsible for influenza outbreaks around the globe in humans [4,5]. Influenza viruses are protected by a lipid bilayer containing the transmembrane proteins hemagglutinin (HA), neuraminidase (NA), and M2. Attached to the inside of the envelope is the matrix protein (M1), which interacts with ribonucleoprotein (RNP) and viral polymerase complexes. The virions can be either spherical or filamentous in structure, with a diameter ranging between 80–120 nm. The viral genome is composed of eight segments of negative-sense single-stranded RNA coding for the different viral proteins [6,7]. The virus takes almost 6 h for its replication in the host cell, ultimately killing the latter [8]. Currently available antiviral drugs to treat influenza virus infection include neuraminidase inhibitors (Oseltamivir, Peramivir, Zanamivir) and polymerase complex PA inhibitors (Baloxavir marboxil) [9,10]. However, while antiviral drugs are relatively easy to manufacture, influenza strains showing resistance to some of them have already been reported [11,12]. 

To date, vaccination remains the most effective means to prevent and contain influenza virus infections. Currently approved vaccines include inactivated influenza vaccine (IIV), live attenuated influenza vaccine (LAIV), and recombinant vaccines (RV), the latter being composed of either subunits or VLPs [13]. IIVs correspond to almost 90% of influenza vaccines produced, and their production relies on an egg-based process developed in the 1940s [14]. Though well-established and cost-effective, egg-based vaccine production is a long process and responsiveness to a potential pandemic scenario heavily relies on egg supply that might be limited. Additionally, some influenza strains such as the H5Nx have shown lower yields when produced in eggs [15,16,17]. The development of cell culture processes for the production of IIVs has gained ground in the last decades, mainly driven by advances in large-scale cell culture techniques. Cell culture-based processes correspond to 20% of the worldwide capacity for pandemic influenza vaccine manufacturing [18]. In short, cells are grown adherent to substrates or in suspension and infected with the desired virus strain, followed by virus harvest, purification, and inactivation. Advantages of this type of process over traditional egg-based production include shorter production cycles, the higher similarity between the vaccine produced and the original strain, and a faster response in the case of a pandemic [15]. Importantly, cell culture-produced influenza vaccines have demonstrated higher efficacy due to better control of the production timelines, hence limiting mismatches in the yearly dominant strain selected [19]. Significant contributions to the acceleration and intensification of the upstream processing of cell-derived influenza candidate vaccines have been completed, however, comparatively limited work has been dedicated to the downstream processing aspects [20,21,22].

In addition to a rapid and robust upstream processing, a rapid and robust purification that can be adapted to the emergence of new strains is particularly important. In recent years, new purification methods based on membrane-based technologies have gained interest in the virus manufacturing industry. Downstream processing starts with the clarification of the harvested material. This step is typically performed by centrifugation at a small scale. However, centrifugation at large-scale often implies a high investment and scale-up limitations, and alternative methods such as filtration are preferred [23]. After its clarification, concentration and purification of the virus are required. This has been traditionally achieved through an ultracentrifugation, performed either with a differential continuous density gradient or with density cushions such as cesium chloride, iodixanol, or sucrose [24,25]. Although established for decades, ultracentrifugation is a capital-intensive investment, with higher equipment maintenance costs and volume limitations. This might be crucial for the different manufactures in pandemic situations. A rapid increase in the production capacities is desired to contain the virus spread. In this context, membrane-based chromatography has emerged as a promising method since it is cost-effective, can be easily scaled-up, can be easily tailored to several influenza viruses, and is compatible with working at high flow rates [26]. In addition, membranes are amenable to continuous manufacturing operations as a trend toward advanced and cost-effective manufacturing strategy [27].

This study focuses on the development of a robust downstream process for the purification of pandemic influenza strains at a bioreactor scale. A two-step purification process consisting of depth filtration followed by anion exchange chromatography is proposed. Depth filtration proved to be a preferred alternative to centrifugation. Then, different membranes including NatriFlo^®^ HD Q, Sartobind^®^ Q, and Mustang^®^ Q were evaluated for the purification of the H1N1 virus as a model of influenza pandemic strains. Successful recovery was demonstrated while establishing the operation conditions. Furthermore, two pandemic strains, H7N9 and H3N2, were purified with the same process, showing great potential for processing different influenza strains for a rapid response to emerging or reemerging influenza strains.

## 2. Materials and Methods

### 2.1. Cell Culture Conditions and Influenza Strains

Productions of influenza viruses were carried out in 1 L and 3 L Applikon benchtop bioreactors, respectively (Table 1). Suspension human embryonic kidney (HEK-293SF) cells were cultured in shake flasks at 37 °C with 5% CO_2_, with serum-free Hycell TransFx (Cytiva, Marlborough, MA, USA) medium supplemented with 4 mM L-Glutamine and 0.1% Kolliphor^®^. Bioreactors were inoculated at cell densities varying from 0.3–0.6 × 10^6^ cells/mL, depending on the operation and feeding strategy. The stirring speed was controlled at 100 rpm, the temperature was maintained at 37 °C using heating blankets, and the pH was controlled at 7.2 using CO_2_ and NaOH. Bioreactors were operated in batch or fed-batch mode, the latter being characterized by the daily addition of 50 mL CellBoost5 (Hyclone) per 1 L culture. Three pandemic strains were produced and analyzed: A/Puerto-Rico/8/34 (H1N1), subtype reassortant carrying the HA and NA genes of the A/Anhui/1/2013 (H7N9), and A/Hong-Kong/8/68 (H3N2). Infection was performed at either low cell density (around 2 × 10^6^ cells/mL) or high cell density (around 7 × 10^6^ cells/mL), at a multiplicity of infection (MOI) ranging from 10^−4^ to 10^−2^, and with the addition of TPCK-trypsin (Sigma, St. Louis, MO, USA) at final concentrations varying from 1–2 µg/mL (both subtype dependent). After infection with the H1N1 subtype, a temperature shift to 35 °C was performed. The produced material was harvested between 48–72 hpi (hours post-infection).

### 2.2. Downstream Processes

A scalable filtration/chromatography-based purification process was performed, which consists of Benzonase^®^ treatment, clarification, inactivation with beta-propiolactone, and ion exchange chromatography in sequence as shown in Figure 1. Both depth filters and ion-exchange chromatographic membranes can be scaled up from laboratory scale to industrial scale by choosing products with larger membrane areas. For clarification, both depth filtration and centrifugation were performed to compare the HA recovery and contamination removal.

#### 2.2.1. Clarification

Benzonase^®^ (Sigma, St. Louis, MO, USA) was added to the bioreactor at 10 units/mL and incubated for 1 h before the clarification step. Depth filtration was evaluated as an alternative to the commonly used centrifugation as a cell removing step. From each harvest, 1 L of the harvested material after Benzonase^®^ treatment was centrifuged at 4000× *g* for 10 min and another 1 L was filtered through MilliStack D0HC^®^ (Millipore, Rockville, MD, USA) depth filter at 7.66 mL/min in sterile conditions as previously described [28,29]. Before filtration, the depth filter was equilibrated with the filter buffer (50 mM HEPES, 300 mM NaCl at pH 7.2). The clarified material was supplemented with HEPES (Sigma, St. Louis, MO, USA) to a final concentration of 70 mM to maintain the pH at 7.2 during the inactivation. The material was inactivated by adding 0.1% (*v*/*v*) β-propiolactone (Sigma, St. Louis, MO, USA) and then stirred for 24 h at 4 °C. The inactivated supernatant was frozen at −80 °C and was thawed upon loading to the AKTA^TM^ Avant 25 system (Cytiva, Marlborough, MA, USA) for evaluation of the ion exchange chromatography membranes.

#### 2.2.2. Ion EXCHANGE chromatography

Inactivated influenza viruses were purified using ion exchange chromatography with an AKTA^TM^ Avant 25 system (Cytiva, Marlborough, MA, USA). Three membranes NatriFlo^®^ HD Q (Sigma, St. Louis, MO, USA), Mustang^®^ Q (Pall, Show Low, AZ, USA), and Sartobind^®^ Q (Sartorius, Göttingen, Germany) were evaluated in the ion exchange chromatography step. Supernatants containing the inactivated viruses were supplemented with 2 mM MgCl_2_ and 0.005% Zwittergent 3-14 (*w*/*v*) before its loading. Ion exchange chromatography was operated in a bind-and-elute mode to separate the virus from residual dsDNA and proteins. UV signals at 260, 280, 290 nm, conductivity, and pH were monitored.

To compare the performance of the ion exchange chromatography using NatriFlo^®^ HD Q with that of the ion exchange chromatography using Sartobind^®^ Q, the flow rate was set at 2 mL/min for all steps. A total of 90 mL of the inactivated supernatant was loaded onto the membrane pre-equilibrated with buffer A (25 mM Tris-HCl, 2 mM MgCl_2_, 0.005% *w*/*v* Zwittergent 3-14, pH 8.1). A wash with 150 mM NaCl, an elution with 1 M NaCl, and a final wash with 2 M NaCl was performed in sequence. The NaCl concentration was controlled by mixing buffer A and buffer B (2.5 M NaCl, 25 mM Tris-HCl, 2 mM MgCl_2_, 0.005% *w*/*v* Zwittergent 3-14, pH 8.1) in proportion. Similarly, to compare the performance of Mustang^®^ Q with that of Sartobind^®^ Q, the flow rate was set at 1 mL/min for all steps. Two-mM MgCl_2_ and 0.005% *w*/*v* Zwittergent 3-14 were added before loading 100 mL of the inactivated supernatant onto each membrane. A wash step with 100 mM NaCl and four elution steps with 300 mM, 700 mM, 900 mM, and 1200 mM NaCl followed by a final wash (regeneration) with 2 M NaCl were performed. For each step, 10 membrane volumes of buffers were used. The NaCl concentration was controlled by mixing buffer A (50 mM HEPES, 2 mM MgCl_2_, 0.005% *w*/*v* Zwittergent 3-14, pH 7.5) and buffer B (2 M NaCl in buffer A, pH 7.5) in proportion. The flow-through, wash, elution, and final wash were collected using the fraction collector. Specifically for the elution step, only the fractions corresponding to UV absorption peaks were collected.

### 2.3. Analysis and Quantification for Impurities and Influenza Viruses

#### 2.3.1. Determination of dsDNA

Double-stranded DNA (dsDNA) was quantified using the PicoGreen^®^ dsDNA quantitation assay kit (Invitrogen, Waltham, MA, USA). Each sample was serially diluted from 1:1 to 1:128 with TE buffer (Invitrogen, Waltham, MA, USA) in Corning™ Polystyrene 96-Well Microplates (Fisher Scientific Inc., Waltham, MA, USA). Duplicated λ dsDNA Standard was diluted with TE buffer from 0 to 500 mg/mL in the same plate. After adding the diluted dye reagent to each well and incubating for 15 min at room temperature, the fluorescence emission was measured at 520 nm with the excitation wavelength at 480 nm (Biotek Instruments, Winooski, VT, USA). The dsDNA concentrations of the samples were quantified using the standard curve. 

#### 2.3.2. Determination of Total Proteins

The total amount of proteins in the samples collected from each step were quantified using the DC protein assay kit (Bio-Rad, Hercules, CA, USA) according to the manufacturer’s protocol. Samples with high concentrations were diluted with Milli-Q water and quantified using the bovine serum albumin standard curve ranging from 0 to 250 μg/mL. Unknown samples and standards were processed in Corning™ Polystyrene 96-Well Microplates (Fisher Scientific Inc., Waltham, MA, USA) and the absorbance at 750 nm was measured using a microplate reader (Biotek Instruments, Winooski, VT, USA).

#### 2.3.3. Dot Blot Assay

The Hemagglutination (HA) concentration was quantified using a Dot-blot protein assay. Previously sucrose cushion purified influenza virus samples with an HA concentration of 25.1 μg/L quantified by a spatial reference identifier (SRID) were diluted to 10 μg/L with phosphate-buffered saline (PBS) as standard. Samples with high concentrations were also diluted with PBS to fit the standard curve. A total of 250 μL of unknown samples and standards were 1:1 mixed with 8 M urea in tris-buffered saline (TBS) (Sigma, St. Louis, MO, USA) and incubated on a 3D rocker (JDM, USA) at room temperature for 30 min. Incubated samples and standards were transferred to a Corning™ Polystyrene 96-Well Microplate in duplicate. Then, the samples were serially diluted 1:1 with 4 M urea for 4 times whereas the standards were serially diluted 1:1 with 4 M urea for 8 times. Immuno-blot polyvinylidene fluoride (PVDF) membranes (Merck Millipore, Rockville, MD, USA) were pre-washed with methanol and Milli-Q water, and then soaked in tris-buffered saline with TBS-Tween before assembling them within the 96-well Bio-Dot microfiltration apparatus that was connected to a vacuum pump. Each well was loaded with 100 μL TBS and slowly drained using a vacuum pump. A total of 100 μL of standards or 100 μL of samples were then transferred to the apparatus accordingly. After the samples were drained, the membrane was removed from the apparatus and incubated in TBS-Tween buffer (containing TBS-Tween and 5% skim milk) with 8 ng/mL universal primary antibody (11H12 from NRC) with shaking for 1 h. After washing the membrane with TBS-Tween buffer 3 times, the membrane was then incubated in TBS-Tween buffer with a secondary antibody (goat-anti-mouse IgG-HRP (H + L), Jackson Immuno-research Labs) at a dilution of 1:15000 with shaking for 1 h. Protein dots on the membrane were visualized with the Clarity™ Western ECL Substrate (Bio-Rad, Hercules, CA, USA) using ChemiDoc (Bio-Rad, Hercules, CA, USA).

#### 2.3.4. Hemagglutination Assay

The hemagglutination assay was performed to quantify the hemagglutinin activity of influenza using chicken red blood cells (RBCs) in 96-v-bottom well microplates (Fisher Scientific Inc., Waltham, MA, USA). RBC concentration was determined and adjusted to 2 × 10^7^ cells/mL. For 96-well flat-bottom microplates, wells from columns 2 to 12 were filled with 100 μL of PBS. Then, 29.3 μL of PBS was added to wells B1, B2, D1, D2, F1, F2, H1, and H2 of each plate. A total of 100 μL of the samples were loaded in the first and second wells of rows A, C, E, and G. A total of 70.7 μL of the samples were added to the first and second wells of rows B, D, F, and H. Starting from column 2, samples were consecutively diluted. A total of 100 μL of RBCs with a concentration of 2 × 10^7^ cells/mL was added to each well. The plates were then incubated at room temperature for 3 h. The value of HA/mL was calculated using the following equation: Influenza viral particle/mL = [RBC] * 10 ^(logHAtiter)^.

## 3. Results

### 3.1. Demonstration of a Scalable Depth Filtration as an Alternative to Centrifugation for H1N1 Clarification

Influenza H1N1 produced from Run #1 and Run #2 with different conditions (Table 1) was subjected to the Benzonase^®^ treatment and clarification process. The initial samples were taken at the time of harvest before Benzonase^®^ treatment. Comparing the samples that were treated with Benzonase^®^ and clarification, Benzonase^®^ treatment shows in both cases a significant effect on dsDNA removal. In the clarification phase, the samples clarified by depth filtration had a better dsDNA removal performance than centrifugation (Table 2). Both methods remove more than 90% of dsDNA, while they have a limited effect on total protein removal. Regarding the HA recovery, depth filtration performed better than centrifugation based on the result of Run #1. From the result of Run #2, both techniques showed similar performance in HA recovery, dsDNA removal, and protein removal since the difference was within 3%. These results indicate that depth filtration is a competitive alternative to centrifugation as a clarification method. The differences in HA recoveries indicate that the higher the HA concentration is, the higher HA recovery can be reached.

### 3.2. Evaluation of Different Membranes for Ion Exchange Chromatography

#### 3.2.1. NatriFlo^®^ HD Q vs. Sartobind^®^ Q

After the depth filtration step, the chromatography step was optimized for the purification of influenza viruses. Chromatography purifications with NatriFlo^®^ HD Q and Sartobind^®^ Q were performed to select the most suitable membrane to purify H1N1 production (Appendix A). As shown in Table 3, for the material generated in Run #3, a fed-batch bioreactor with a significantly higher dsDNA accumulation as compared to the starting material from Run #1 and Run #2, the dsDNA removal in the clarification step was still around 90%, indicating a good performance of Benzonase^®^ treatment. At 1 M salt concentration, the Sartobind^®^ Q had 70.7% HA recovery, which is 28.7% higher than that of NatriFlo^®^ HD Q, indicating a higher binding capacity of Sartobind^®^ Q. In terms of impurities removal, eluate from NatriFlo^®^ HD Q had a 35.3% DNA removal which is 24.5% higher than that from Sartobind^®^ Q. For proteins removal, NatriFlo^®^ HD Q had a better performance than Sartobind^®^ Q with 83.8% and 74.9%, respectively. NatriFlo^®^ HD Q performed better for impurities removal than Sartobind^®^ Q. Based on the comparison results of the NatriFlo^®^ HD Q and Sartobind^®^ Q, the latter was selected for further purification process development due to significantly better performance in HA recovery.

#### 3.2.2. Sartobind^®^ Q vs. Mustang^®^ Q

Mustang^®^ Q has been extensively used as a membrane chromatography for the purification of different viral products [30,31,32]. Therefore, the performance of Mustang^®^ Q was evaluated and compared with Sartobind^®^ Q. The starting material for this set of experiments was obtained from a depth filtration of Run #1 harvest material (Table 1). During loading of the clarified supernatant to the columns, both dsDNA and virus are expected to bind to the column as they are negatively charged. Clear UV peaks are observed during the elution step as shown in Appendix A. For mustang^®^ Q, UV peaks were detected during all four elution steps in a downward trend and also during the final wash, while for Sartobind^®^ Q, UV peaks were only visible in the first three elutions.

Assay results of cumulative HA recovery, dsDNA, and protein removal during the chromatography step are shown in Figure 2. The total recovery values of HA from Mustang^®^ Q and Sartobind^®^ Q were 54.2% and 65.3%, respectively. When using Mustang^®^ Q, the eluent from 900 mM and 1200 mM NaCl contains 8.2% HA but 67.3% dsDNA. Thus, these fractions were not collected due to the consideration of dsDNA removal. When using Sartobind^®^ Q, the eluent from 900 mM NaCl contains 3.0% HA and 17.2% dsDNA. Although there is 70.1% dsDNA in 700 mM NaCl fraction of Sartobind^®^ Q, the eluent was collected because 43.4% HA was recovered. Overall, to balance the HA recovery and dsDNA removal, the fractions eluted at 300 mM and 700 mM were collected and mixed and the result is shown in Table 4. The HA recovery of Sartobind^®^ Q is 62.5% which is 16.7% higher than that of Mustang^®^ Q membrane. For the protein removal, Mustang^®^ Q has a better performance than Sartobind^®^ Q, since 75.0% of proteins were washed out in flow through fraction, which is 11.2% higher than for Sartobind^®^ Q.

The targeted concentration of HA should be in the range of 8–15 μg/dose, while dsDNA should be less than 10 ng/dose, and the ratio of protein versus HA should be less than 6 to meet the requirement of animal tests [33,34]. As shown in Table 4, when there is 15 μg HA in one dose, the product from Mustang^®^ Q contains DNA at a concentration of 19.6 ng per dose which is lower than 41.1 ng per dose for Sartobind^®^ Q. Proteins per HA from Mustang^®^ Q are still lower than Sartobind^®^ Q. The performance of HA recovery and impurities removal on the purification of H1N1 shows different characteristics of Mustang^®^ Q and Sartobind^®^ Q.

### 3.3. Evaluation of Clarification and Membrane Chromatography Steps for Purification of Pandemic Influenza Strains H7N9 and H3N2

#### 3.3.1. Clarification for Influenza H7N9 and H3N2

Two potential influenza pandemic strains H7N9 and H3N2 were produced in Run #4-#7. A series of experiments were performed to evaluate the performance of depth filtration versus centrifugation, and the results are summarized in Table 5. Over 80% of host cell DNA and 16% of proteins were removed by clarification methods of depth filtration and centrifugation, while 70% of HA were recovered in both cases. When measuring HA recovery, HA assay was used since the primary and secondary antibodies were specific to H1N1. The results in the case of H1N1 show that having a higher initial concentration of HA produces a higher recovery, which varies in the case of H3N2, in which higher recoveries are found independently of the initial HAU amounts. The different strains may have accounted for these differences. These results indicated that both depth filtration and centrifugation methods show similar performances for the clarification step of H7N9 and H3N2 cell culture harvest material. 

#### 3.3.2. Implement of Ion Exchange Chromatography for Influenza H7N9 and H3N2

H7N9 and H3N2 material harvested from bioreactor productions Run #4 and #6 were purified by the Mustang^®^ Q membrane (Table 6). After the loading step, the viral particles were eluted with a step gradient elution mode by 0.3 M and 0.7 M NaCl. HA assays were performed to evaluate the recovery of the viruses. Results showed that the HA recoveries of H7N9 and H3N2 were 41.6% and 41.3%. Similar impurities removal performance was achieved with H7N9 and H3N2, as more than 90% proteins and about 40% dsDNA were removed by Mustang^®^ Q. These results demonstrate that the ion exchange membrane chromatography downstream processing step is a promising method to purify various influenza strains.

## 4. Discussion

With seasonal influenza circulating worldwide and new influenza strains constantly emerging, it is urgent to develop a generic and cost-efficient influenza vaccine manufacturing platform. In the upstream process, the influenza strains including H1N1, H3N2, and H7N9 were produced by HEK-293SF in suspension cultures. Compared with cultivation in embryonated eggs, manufacturing of cell culture-derived influenza vaccines is a rapidly growing field, which is promising to realize a robust, scalable, and high-yield production process to meet the global requirements [20,21,35]. Additionally, since a large portion of the process development costs is related to downstream steps, process intensification for these is worth considering to reduce the cost per dose [23,36]. In this study, a membrane chromatography-based downstream process including DNA treatment, clarification, inactivation, and ion exchange chromatography was developed and optimized. Meanwhile, culture materials harvested from bioreactor batches with different strains and viral productions were collected and used as initial materials under different conditions to test the robustness of the downstream process steps. Overall, we show that the chromatography-based proposed downstream process builds up a generic platform to purify the cell culture-produced influenza vaccine candidates and provides a reference for the purification of other enveloped viruses.

In the clarification step, material harvested from two bioreactor batches with a different accumulation of impurities and HA concentration were processed to evaluate the performances of depth filtration against centrifugation for HA recovery and impurities removal starting from different initial materials. Influenza H1N1 production performed in 3 L Run #2 used a different MOI leading to a higher cell density as compared to material generated in Run #1 (Table 1). The clarification step of depth filtration is compared with centrifugation focusing on HA recovery and impurities (total proteins, dsDNA) removal. Overall, depth filtration and centrifugation have similar performance considering impurities removal and viral recovery. With the consideration of large-scale production, depth filtration is an ideal technology for industrial manufacturing of these influenza strains. Additionally, depth filtration contributes to a sterile clarification since the viral material can be directly transferred to the depth filter from the bioreactor.

The separation step plays an important role in purifying influenza material. There are two categories of technologies involved. One is the chromatographic method which includes membrane-based and packed bed-based chromatography. The other is the ultracentrifugation method. Traditionally, the viral material of the influenza vaccine is produced from chicken embryos and purified by zonal gradient centrifugation [37,38]. Asanzhanova et al. implemented the purification of influenza virus strain A/NIBRG-121xp, N1H1 by ultracentrifugation and chromatographic methods [39]. The HA recoveries achieved with these methods were 61.7% and 54.5%, respectively. Although zonal ultracentrifugation provides a higher HA recovery, the process is more time-consuming than chromatography. The pilot ultracentrifugation process is preceded by pre-treatment for 8–12 h while the chromatographic method can be performed within 6–10 h, which indicates that chromatography is a promising alternative to traditional sucrose gradient zonal ultracentrifugation. Except for membrane-based chromatography, the packed bed-based chromatography is another downstream process step developed for influenza vaccine purification. Tseng et al. used in their study a CaptoCore 700 pre-packed column for purification of influenza H7N9. They reported 33–46% HA recovery yields by this platform [26]. Weigel et al. developed a flow-through chromatographic process for purification of Influenza A and B. There were three steps involved, including ion exchange chromatography, Benzonase^®^ treatment, and size exclusion chromatography [40]. However, packed beds are composed of beads packed into a column. Several disadvantages associated with this type of chromatographic column have been reported, including limitations of operation flow rate, low dynamic binding capacity, and necessary compromise between pressure drop and mass transfer resistances [23,36,41].

With the consideration of efficiency, a membrane chromatography-based operational unit was selected for the purification of H1N1. Three membranes from different vendors including NatrixFlo^®^ HD Q, Mustang^®^ Q, and Sartobind^®^ Q were evaluated according to the performance of HA recovery and impurities removal. The comparison experiment was started from the purification of H1N1 by NatriFlo^®^ HD Q and Sartobind^®^ Q. Since the latter shows a significantly better performance in HA recovery than NatriFlo^®^ HD Q, Sartobind^®^ Q was selected for further purification process development. In the next step of the comparison experiment, Sartobind^®^ Q and Mustang^®^ Q were evaluated. In this case, a purification process with a gradient using more steps was performed. After balancing the HA recovery and impurities removal, eluents from 300 mM to 700 mM NaCl were collected. The results show the different performances of Sartobind^®^ Q and Mustang^®^ Q, which provides the rationale for selecting the membrane. Mustang^®^ Q is good at impurities removal whereas Sartobind^®^ Q shows better performance in viral recovery. All parameters of the purified H1N1 were adapted to the standard format to meet the requirement for animal experiments. The targeted concentration of HA should be 8–15 μg/dose. Regarding the contamination removal, the dsDNA should be less than 10 ng/dose, and the ratio of protein versus HA should be less than 6. If there is 8 μg of HA in one dose, influenza purified by Mustang^®^ Q meets all the quality requirements with an HA recovery of 45.8%. When using Sartobind^®^ Q, although the HA recovery could reach 62.5%, a polishing step might be required because of the excess of DNA per dose. When 15 μg of HA in one dose is required, for both membranes a polishing step might be required. In this case, Sartobind^®^ Q would be the optimal solution because of the high HA recovery.

To generalize the application scope and establish a generic platform for downstream processing of pandemic influenza strains, the comparison of depth filtration and centrifugation for H7N9 and H3N2, two potential influenza pandemic strains, was firstly performed based on the same clarification protocols as described in Section 3.1. As shown by the results presented in Table 5, both methods have similar performances for the clarification step of H7N9 and H3N2 cell culture harvest material. After the clarification step, the same downstream processing steps under similar operating conditions were applied to the purification of H7N9 and H3N2 material. For both strains, recovery of more than 40% of HA and similar impurities removal performances were achieved similarly to H1N1 results. Overall, this data supports the generalization of the findings established with the H1N1 purification process and suggest that the chromatographic membrane-based downstream processing is a promising technology for the purification of different influenza strains.

## 5. Conclusions

In this study, a chromatographic membrane downstream process was evaluated for purification of influenza strains and further developed and optimized. With the aim to streamline the large-scale H1N1 influenza strain purification process, first-depth filtration was compared to centrifugation. The depth filtration achieved a similar performance compared to the centrifugation as a clarification method. Second, a membrane chromatography step was performed to separate the influenza viruses from contaminants. Three commercial ion exchange membranes including NatriFlo^®^ HD-Q, Mustang^®^ Q, and Sartobind^®^ Q were evaluated. Sartobind^®^ Q showed higher HA recovery despite lower impurities removal than NatrixFlo^®^ HD-Q. When comparing Sartobind^®^ Q and Mustang^®^ Q, Sartobind^®^ Q had better performance in HA recovery, whereas Mustang^®^ Q was better in impurity removal leading to the selection of Mustang^®^ Q for further validation. The optimized operating purification conditions were further evaluated with two pandemic strains H7N9 and H3N2, resulting in comparable downstream process performances than H1N1. Overall, these results broaden the application range of the membrane chromatography-based purification process for influenza vaccines and provide evidence for a cost-effective alternative to ultracentrifugation technologies currently used in influenza vaccine manufacturing. Both depth-filtration for clarification and ion-exchange chromatographic membranes are well-established technologies for rapid deployment in existing manufacturing capacity for rapid response to influenza pandemic surge situations.

## Figures and Tables

**Figure 1 vaccines-10-01310-f001:**
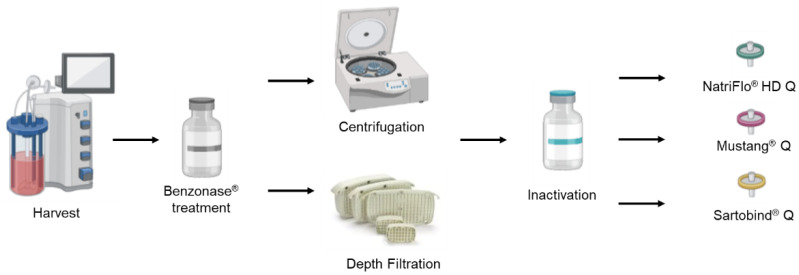
Downstream processing flow and alternative clarification steps and ion exchange chromatography steps compared in the study.

**Figure 2 vaccines-10-01310-f002:**
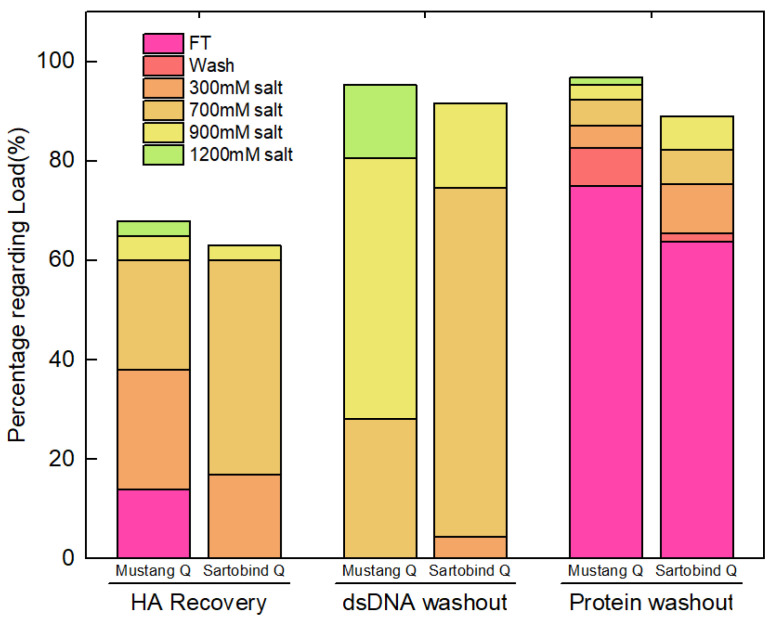
Assay results of chromatography for Mustang^®^ Q and Sartobind^®^ Q from Run #1. The percentages are calculated regarding loading material. Each fraction is shown with a different color. The first, second, and third two bars represent dsDNA, HA recovery, and proteins, respectively.

**Table 1 vaccines-10-01310-t001:** Upstream operational conditions and quantification.

No.	Strain	Bioreactor Volume	Cell Density at Infection	MOI	TPCK-Trypsin	Temperature at Infection	Time of Harvest	Total Cell Density at Harvest	Cell Viability at Harvest
Run #1	A/PR/8/34 H1N1	2.3 L	1.3 × 10^6^ cells/mL	0.001	1 μg/mL	35 °C	45 hpi	1.1 × 10^6^ cells/mL	26%
Run #2	A/PR/8/34 H1N1	2.3 L	1.4 × 10^6^ cells/mL	0.01	1 μg/mL	35 °C	40 hpi	4.7 × 10^6^ cells/mL	63%
Run #3	A/PR/8/34 H1N1	750 mL	7.5 × 10^6^ cells/mL	0.001	1 μg/mL	35 °C	48 hpi	4.0 × 10^6^ cells/mL	58%
Run #4	A/Anhui/1/2013 H7N9	2.3 L	6.5 × 10^6^ cells/mL	0.0001	1 μg/mL	37 °C	54 hpi	3.5 × 10^6^ cells/mL	88%
Run #5	A/Anhui/1/2013 H7N9	2.3 L	1.9 × 10^6^ cells/mL	0.0001	1 μg/mL	37 °C	56 hpi	3.6 × 10^6^ cells/mL	53%
Run #6	A/Hong-Kong/8/64 H3N2	2.3 L	1.4 × 10^6^ cells/mL	0.01	2 μg/mL	37 °C	96 hpi	1.3 × 10^6^ cells/mL	72%
Run #7	A/Hong-Kong/8/64 H3N2	750 mL	1.9 × 10^6^ cells/mL	0.01	2 μg/mL	37 °C	48 hpi	3.4 × 10^6^ cells/mL	66%

**Table 2 vaccines-10-01310-t002:** Comparison of clarification methods between centrifugation and depth filtration.

No.	Sample ID	HA (μg/mL)	DNA (ng/mL)	Protein (μg/mL)	HA Recovery	DNA Removal	Protein Removal
Run #1	Initial	130.9 ± 17.3	7748.0 ± 89.7	262.2 ± 14.8			
	Depth filtration	103.6 ± 15.1	268.5 ± 17.0	190.3 ± 16.4	79.1 ± 11.5%	96.5 ± 6.1%	27.4 ± 2.4%
	Centrifugation	92.2 ± 10.4	636.3 ± 39.2	179.8 ± 9.5	70.3 ± 7.9%	91.8 ± 5.7%	31.4 ± 1.7%
Run #2	Initial	28.5 ± 3.5	28,229.1 ± 73.9	402.7 ± 28.4			
	Depth filtration	10.9 ± 1.25	1247.2 ± 27.2	272.3 ± 16.8	35.4 ± 4.1%	95.6 ± 9.0%	32.4 ± 2.0%
	Centrifugation	10.4 ± 0.8	2012.0 ± 38.1	280.7 ± 18.2	36.5 ± 2.8%	92.9 ± 8.7%	30.3 ± 2.0%

**Table 3 vaccines-10-01310-t003:** Chromatography results of NatriFlo^®^ HD Q vs. Sartobind^®^ Q.

No.	Sample ID	Volume (mL)	HA (μg/mL)	DNA (ng/mL)	Protein (μg/mL)	HA Recovery	DNA Removal	Protein Removal
Run #3	Initial	124.3	18.9 ± 1.8	77,237.3 ± 734.6	786.0 ± 30.6			
	Load	112.7	12.6 ± 0.7	8170.9 ± 162.3	698.0 ± 27.3	60.4 ± 3.3%	90.4 ± 7.3%	19.5 ± 1.8%
	NatriFlo^®^-1 M	11.4	52.3 ± 2.8	52,289.7 ± 525.9	1116.6 ± 35.4	42.0 ± 3.8%	35.3 ± 2.7%	83.8 ± 6.3%
Run #3	Initial	128.4	18.9 ± 1.8	77,237.3 ± 734.6	786.0 ± 28.5			
	Load	122.1	14.1 ± 1.1	7400.6 ± 154.7	737.3 ± 26.1	70.9 ± 5.3%	90.9 ± 7.1%	10.8 ± 0.8%
	Sartobind^®^ Q-1 M	16.4	74.2 ± 4.7	49,169.3 ± 502.0	1377.6 ± 38.9	70.7 ± 4.5%	10.8 ± 0.8%	74.9 ± 7.0%

**Table 4 vaccines-10-01310-t004:** Different performance of Mustang^®^ Q and Sartobind^®^ Q in ion exchange chromatography step using the supernatant from Run #1.

No.	Sample ID	Volume (mL)	HA (μg/mL)	DNA (ng/mL)	Protein (μg/mL)	HA Recovery	ng DNA/Dose	Protein /HA
Run #1	Load	100.0	37.5 ± 2.6	80.5 ± 5.0	193.5 ± 11.4			
	Mustang^®^ Q	17.0	101.0 ± 7.6	132.2 ± 10.7	130.9 ± 7.9	45.8 ± 3.5%	19.6 ± 1.6	1.3 ± 0.1
	Sartorbind^®^ Q	37.5	62.5 ± 3.2	171.1 ± 9.1	109.1 ± 8.9	62.5 ± 3.2%	41.1 ± 2.2	1.7 ± 0.1

**Table 5 vaccines-10-01310-t005:** Comparison of clarification methods between centrifugation and depth filtration.

No.	Sample ID	HAU (unit/mL)	DNA (ng/mL)	Protein (μg/mL)	HAU Recovery	DNA Removal	Protein Removal
Run #4	Initial	1230.5 ± 85.8	2793.6 ± 67.2	125.1 ± 11.5			
H7N9	Depth filtration	867.4 ± 65.3	196.1 ± 10.7	104.8 ± 9.9	70.5 ± 0.4%	93.0 ± 5.1%	16.2 ± 1.5%
Run #5	Initial	888.4 ± 81.3	9276.8 ± 185.0	202.7 ± 20.0			
H7N9	Centrifugation	641.1 ± 65.3	1867.1 ± 53.7	168.8 ± 11.6	72.9 ± 13.1%	79.9 ± 6.5%	16.7 ± 1.2%
Run #6	Initial	28.9 ± 1.0	2341.6 ± 53.6	251.7 ± 22.6			
H3N2	Depth filtration	22.8 ± 4.8	275.4 ± 12.3	200.6 ± 16.3	79.1 ± 18.2%	88.2 ± 6.3%	20.3 ± 1.7%
Run #7	Initial	163.2 ± 5.6	4441.4 ± 89.1	113.5 ± 10.6			
H3N2	Centrifugation	117.7 ± 4.0	248.4 ± 11.2	90.0 ± 7.8	72.1 ± 2.4%	94.4 ± 8.4%	20.7 ± 1.8%

**Table 6 vaccines-10-01310-t006:** Purification of pandemic strains H7N9 and H3N2 with Mustang^®^ Q.

No.	Sample ID	Volume (mL)	HAU (unit/mL)	DNA (ng/mL)	Protein (μg/mL)	HAU Recovery	DNA Residual	Protein Residual
Run #4	Load	147.5	797.3 ± 4.6	185.3 ± 17.9	103.2 ± 5.2			
H7N9	0.3 M	7.0	1771.4 ± 171.4	79.7 ± 4.9	42.5 ± 3.0	10.5 ± 1.0%	2.0 ± 0.1%	1.9 ± 0.1%
	0.7 M	7.0	5223.5 ± 179.3	2175.8 ± 195.8	49.6 ± 3.1	31.1 ± 1.0%	55.7 ± 5.0%	2.3 ± 0.1%
				Total HAU Recovery: 41.6%				
Run #6	Load	100	22.8 ± 4.8	174.4 ± 17.0	202.9 ± 18.1			
H3N2	0.3 M	8.5	29.4 ± 1.0	21.2 ± 1.3	131.9 ± 12.8	11.3 ± 2.5%	1.0 ± 0.1%	5.5 ± 0.5%
	0.7 M	7.5	89.6 ± 8.2	1383.8 ± 102.5	82.2 ± 6.0	30.0 ± 3.2%	59.5 ± 4.4%	3.0 ± 0.2%
				Total HAU Recovery: 41.3%				

## Data Availability

Data will be made available upon request.

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
