# Peer review of "Membrane Chromatography-Based Downstream Processing for Cell-Culture Produced Influenza Vaccines"

_vaccines, 2022, doi:10.3390/vaccines10081310_

Round 1

Reviewer 1 Report

the manuscript the trial of purifying influenza viruses with new strategies (depth filtration and SEC) and comparing the results with low speed centrifugation and ultracentrifugation. The new methodology is dependent on removing cell debris by depth filtration followed by benzonase treatment to get rid of the cellular DNA, inactivation and a final step of purification with SEC. It seems that the new strategy revealed a result comparable and better than the ordinary centrifugation methods.

The manuscript is well-written and the results are well-presented.

It would be important if the authors shed light on the possibility to apply this method to other viruses as well.

Can the authors describe to what volume scale can this method be applied?

There are some viruses that are hard to be grown on cell culture and using embryonated eggs is necessary. can this method be modified to suit  viruses harvested after egg infection?

Author Response

Reviewer: 1

The manuscript the trial of purifying influenza viruses with new strategies (depth filtration and SEC) and comparing the results with low speed centrifugation and ultracentrifugation. The new methodology is dependent on removing cell debris by depth filtration followed by benzonase treatment to get rid of the cellular DNA, inactivation and a final step of purification with SEC. It seems that the new strategy revealed a result comparable and better than the ordinary centrifugation methods.

Response to Reviewer 1:

Thank you for the time you dedicated to revise the manuscript. We highly appreciate your comments and suggestions on the manuscript. Based on those comments, we have revised the manuscript accordingly. The changes are marked using the “Track Changes” function in the revised version of manuscript.

  1. It would be important if the authors shed light on the possibility to apply this method to other viruses as well.

Response:

Thank you for this comment. The two-step purification process consisting of depth filtration and anion exchange chromatography is a promising strategy for purifying enveloped viruses. In this study, we demonstrated the advantages of using the strategy to purify three different strains including H1N1, H3N2, and H7N9. As mentioned in the Discussion section, the proposed membrane chromatography purification is a scalable and generic method for the processing of different influenza strains. Indeed, there is work in progress in our Lab in which we are applying this approach (clarification and membrane-based chromatography for purification) to other enveloped viruses or envelope viral vectors with encouraging results. These results are being compiled and will be the subject of future work. We have added a sentence as “The chromatography-based proposed downstream process provides a reference for the purification of other enveloped viruses.” (Page 10 Lines 345-347)

  1. Can the authors describe to what volume scale can this method be applied?

Response:

The developed downstream process is promising to be used on different scales from laboratory to industry. In the depth filtration step, we used the MilliStack D0HC® with a membrane area of 23 cm2 from the Millipore. For larger scale clarification, as much as 1.1 m2surface area of the filter is available to apply. Regarding the ion-exchange chromatography step, both Mustang® Q (Pall, USA), and Sartobind® Q (Sartorius, Germany) provide capsule formats for pilot or industrial scale applications. We have added expression as “Both depth filters and ion-exchange chromatographic membranes can be scaled up from laboratory scale to industrial scale by choosing products with larger membrane areas.” (Page 3 Lines 129-131)

  1. There are some viruses that are hard to be grown on cell culture and using embryonated eggs is necessary. Can this method be modified to suit viruses harvested after egg infection?

Response:

We agree with the reviewer that many vaccines including influenza are still produced in eggs. Consequently, It might be important to adapt the method for embryonated egg-based vaccine manufacturing. For egg-based virus purification, the rheology of the fluid brings a challenge to clarification. Blom et al. performed a clarification method by continuous disk stack centrifugation and filtration using two parallel 10 in. 2.0 μm ULTA™ prime capsule filters, which would be still advantageous over ultracentrifugation [1]. The clarified viral materials can be further purified by our proposed membrane chromatography-based downstream process. This modified strategy could serve as an alternative to the traditional sucrose zonal gradient ultracentrifugation when purifying certain viruses from eggs. Additional experiments would be needed to demonstrate it.

Reference:

[1] Blom, H.; Akerblom, A.; Kon, T.; Shaker, S.; van der Pol, L.; Lundgren, M. Vaccine 2014, 32, 3721-3724, doi:10.1016/j.vaccine.2014.04.033.

Reviewer 2 Report

In this study, the authors investigated a method using membrane chromatography for purification of viruses in the manufacturing process of influenza vaccines. First, the authors compared clarification by the centrifugal method and the depth filtration membrane, and clarified that the depth filtration membrane showed the same performance as the centrifugation method. It was also shown that Mustang Q is superior in impurity removal in the membrane chromatography process. In addition, optimized purification conditions were also evaluated with H7N9 and H3N2 viruses, yielding comparable performance to H1N1. These results will contribute to the development of cost-effective alternatives to current ultracentrifugation techniques and to rapid vaccine production.

There are some comments about the content of this paper.

1.  The authors explain the results in Table 2 as “The differences of HA recoveries indicates that the higher the HA concentration is, the higher HA recovery can be reached”. On the other hand, #6 in Table 5 corresponds to #2 in Table 2, and #7 is considered to correspond to #1, but these results indicate as “Over 80% ~~, while 70% of HA were recovered in both cases”. The authors need to explain these discrepancies.

2.  Tables 2, 3 and 4 are shown in HA (ug/ml) and Tables 5 and 6 are shown in HAU (unit/ml). Couldn't these units be displayed in either?

3.  HAU recovery in Table 6 is listed separately for 0.3 M and 0.7 M, but the total value is listed in the text (p.9 lane 321). Can these total values also be displayed in Table 6?

4.  Samples #4 and #6 in Table 6 differed by 2-fold in protein amount and greatly differed in HAU/Protein, but the HA recovery value and ratio of residual DNA and protein in each fraction were similar. Is it correct to assume that this method cannot specifically purify viruses? Also, if these fractions are analyzed by SDS-PAGE, will a similar band pattern be detected?

Author Response

Reviewer: 2

In this study, the authors investigated a method using membrane chromatography for purification of viruses in the manufacturing process of influenza vaccines. First, the authors compared clarification by the centrifugal method and the depth filtration membrane, and clarified that the depth filtration membrane showed the same performance as the centrifugation method. It was also shown that Mustang Q is superior in impurity removal in the membrane chromatography process. In addition, optimized purification conditions were also evaluated with H7N9 and H3N2 viruses, yielding comparable performance to H1N1. These results will contribute to the development of cost-effective alternatives to current ultracentrifugation techniques and to rapid vaccine production.

Response to Reviewer 2:

Thank you for the time you dedicated reviewing our manuscript. We highly appreciate your comments and suggestions on our manuscript. Based on those comments, we have revised our paper accordingly. The changes are marked using the “Track Changes” function in the manuscript.

Comments/questions:

  1. The authors explain the results in Table 2 as “The differences of HA recoveries indicates that the higher the HA concentration is, the higher HA recovery can be reached”. On the other hand, #6 in Table 5 corresponds to #2 in Table 2, and #7 is considered to correspond to #1, but these results indicate as “Over 80% ~~, while 70% of HA were recovered in both cases”. The authors need to explain these discrepancies.

Response:

We agree that in the case of the H3N2 strain, higher initial recovery of HA was not needed in order to obtain higher recoveries. Therefore, we modified the sentence in 3.1. and in the revised version, we mentioned that “The results in the case of H1N1 show that having a higher initial concentration of HA produces a higher recovery, which varies in the case of H3N2, in which higher recoveries are found independently of the initial HAU amounts. The different strains may have accounted for these differences.” (Page 9 Lines 311-314). The main observation is that different clarification methods including centrifugation and depth filtration rendered equivalent results in terms of recovery, which demonstrates that depth filtration would be a preferred alternative to centrifugation avoiding high capital investment at large scale.

  1. Tables 2, 3 and 4 are shown in HA (ug/ml) and Tables 5 and 6 are shown in HAU (unit/ml). Couldn't these units be displayed in either?

Response:

Thank you for your question. In the downstream processing for H1N1, the Hemagglutination (HA) concentration was quantified using the Dot-blot protein assay in which the primary antibody (11H12 from the National Research Council of Canada) and secondary antibody (goat-anti-mouse IgG-HRP (H+L), Jackson Immuno-research Labs) were used. And then, we extended the application of membrane chromatography-based downstream processing to two pandemic strains H3N2 and H7N9 to evaluate the generic of the purification strategy. We used Hemagglutination Assay to quantify the HA of H3N2 and H7N9 since the antibodies were specifically used for H1N1. The encouraging results broaden the application range for influenza vaccines and provide evidence for a cost-effective alternative to ultracentrifugation technologies currently used in influenza vaccine manufacturing.

  1. HAU recovery in Table 6 is listed separately for 0.3 M and 0.7 M, but the total value is listed in the text (p.9 lane 321). Can these total values also be displayed in Table 6?

Response:

Thank you for your suggestion. We have revised the table accordingly. (Page 10 Line 329)

  1. Samples #4 and #6 in Table 6 differed by 2-fold in protein amount and greatly differed in HAU/Protein, but the HA recovery value and ratio of residual DNA and protein in each fraction were similar. Is it correct to assume that this method cannot specifically purify viruses? Also, if these fractions are analyzed by SDS-PAGE, will a similar band pattern be detected?

Response:

The developed downstream process method was able to render around 41% recovery independently of the initial amount of HAU. In the case of H7N9, with 117,601 units of HAU, the recovery was 41.6%, while in the case of H3N2, with an initial amount of 2280 units of HAU, the recovery was 41.3%. The results showed a similar and specific purification capacity for both strains. Regarding the proteins, the method was able to eliminate about 95% of contaminant proteins independently of the initial amount of total proteins. Similarly, it was successful in eliminating around 40% of contaminant DNA. Overall, the data showed an acceptable ability to discard an important amount of contaminants while having an HAU recovery of 41%. Regarding the fractions being analyzed in SDS-PAGE, we would expect similar patterns of contaminants since the two bioreactors were run using HEK293SF cells as producers of the different influenza strains. For us, an important fact was the ability to decrease over 90% the presence of contaminant proteins in the final virus elution.